# Multi-Marginal $f$-divergence Schrödinger Bridges: Towards A Unifying Framework for Generation and Distillation

## Abstract

We propose a unified framework for multimodal generation and knowledge distillation by leveraging the Multi-marginal Static Schrödinger Bridge (MSSB) with general $f$-divergence, where we use flexible and task-oriented prior measures. This approach allows us to adapt the MSSB problem to diverse tasks—from text-guided image generation to model compression—simply by designing an appropriate prior. For generative modeling, we develop an efficient block-stochastic optimization scheme and a practical Langevin-based inference method. For knowledge distillation, this framework has a clear information-theoretic interpretation: we prove that our MSSB-based Knowledge Distillation (MSSB-KD) implements a variational relaxation of the Information Bottleneck principle. Our novel MSSB-KD formulation demonstrates strong robustness to noisy supervision, significant gains in multi-teacher settings, and scalability across architectures. Finally, we theoretically prove the equivalence between Static and Dynamic Schrödinger Bridges for general $f$-divergences, enabling the use of divergences better suited to the task at hand.

## 1 Introduction

The Schrödinger Bridge (SB) framework has emerged as a powerful tool for generative modeling, optimal transport, and stochastic control. By seeking the most likely stochastic process interpolating between two marginals under entropy regularization, SB methods have achieved success in image generation, diffusion modeling, and trajectory inference. However, prior SB approaches are restricted to *two marginals*, preventing their application to multimodal tasks that require modeling relationships between three or more modalities—a critical capability for modern applications involving image, text, audio, and other domains.

We present a practical extension of the Static Schrödinger Bridge (SSB) framework to the *multi-marginal* setting, unlocking its potential for multimodal generation and knowledge distillation. As an example of multimodal generation, for text-guided image generation, we model the joint relationship between an input image $x$, text instruction $t$, and output image $y$ as a three-way SSB problem. This formulation learns an entropy-regularized transport plan that simultaneously ensures semantic alignment with instructions, preservation of source structure, and adherence to realistic output distributions.

While most existing SB-based generative modeling methods employ the Dynamic Schrödinger Bridge (DSB) formulation Bortoli et al. (2023); Liu et al. (2023); Chen et al. (2021); Vargas et al. (2021), we adopt the Static Schrödinger Bridge (SSB) framework instead. This choice offers several key advantages: (i) computational efficiency through learning a direct coupling rather than an entire stochastic process trajectory, (ii) elimination of the need to sample from stochastic differential equations during training, (iii) flexibility in choosing the prior measure through customizable cost functions, (iv) freedom in parameterizing the dual potentials using arbitrary neural network architectures, and (v) efficient block optimization through the dual formulation that enables scalable training. These advantages make SSB particularly well-suited for multi-marginal problems where the computational complexity of DSB would be prohibitive.

Our Multi-marginal Static Schrödinger Bridge (MSSB) problem with a Gibbs prior is equivalent to an entropic optimal transport (EOT) formulation, yielding an optimal transport plan that takes the form

of a Gibbs measure as well. We incorporate textual semantics through cross-attention and design latent-space cost functions that preserve perceptual and identity constraints. The dual formulation over potentials $(\alpha, \beta, \gamma)$ admits efficient block-coordinate optimization with convergence guarantees, while maximizing a variational lower bound on conditional mutual information $I(Y; T|X)$.

This framework naturally extends to knowledge distillation by treating data, teacher logits, and student logits as three marginals. We prove that MSSB-based knowledge distillation (MSSB-KD) implements a variational relaxation of the Information Bottleneck (IB) principle, establishing a rigorous connection between optimal transport and information-theoretic compression.

Lastly, a key theoretical advance we make is the proof that Dynamic and Static Schrödinger Bridges are equivalent for *any f-divergence*, not just KL divergence. This generalization is significant: different $f$-divergences emphasize different aspects of the transport problem—Total Variation for sharp boundaries, $\chi^2$ for outlier robustness, and $\alpha$-divergences for interpolating between mode-seeking and mean-seeking behavior. By establishing this equivalence, we enable practitioners to choose the divergence that best matches their application requirements. For image translation tasks, this means we can select divergences that preserve fine details (using $\alpha < 1$) or ensure smooth transitions (using $\alpha > 1$), opening the door to learnable, task-specific divergences in future work.

**Contributions.** We summarize our contributions as below

- We introduce a unified MSSB framework for multimodal generation and knowledge distillation, whereas most existing works use the more computationally heavy DSB.
- We prove equivalence between dynamic and static formulations for general $f$-divergences, going beyond KL to enable application-specific divergence selection and paving the way for learnable divergence functions that adapt to task requirements.
- We show that MSSB-KD recovers an Information Bottleneck-style trade-off b/w compression and fidelity.
- We develop efficient block-stochastic optimization and Langevin-based inference algorithms for MSSB.
- We empirically validate our approach on multimodal generation and distillation tasks. For distillation, we demonstrate robustness to noisy supervision, successful multi-teacher distillation, and consistent improvements across diverse architectures.

## 2 RELATED WORK

**Schrödinger Bridges and Optimal Transport.** The Schrödinger Bridge (SB) problem, originally formulated by Schrödinger (1931; 1932), seeks the most likely stochastic evolution between two probability distributions. Föllmer (1988) formalized this as a relative entropy minimization problem, establishing connections to time-reversal of diffusion processes. Léonard (2013a;b) provides comprehensive surveys connecting Schrödinger bridges to optimal transport theory.

The modern optimal transport framework, pioneered by Kantorovich (1942) and Monge (1781), has seen renewed interest through computational advances. The entropic regularization approach, popularized by Cuturi, enables efficient computation via the Sinkhorn algorithm Sinkhorn (1964); Knopp & Sinkhorn (1967).

Recent work has revitalized SB methods for ML. Chen et al. (2021) developed likelihood training for SB. Bortoli et al. (2023) introduced Diffusion Schrödinger Bridge (DSB), combining score-based models with SB theory. Vargas et al. (2021) proposed iterative proportional fitting for solving SB problems. Korotin et al. (2024) introduced Light Schrödinger Bridge for direct static formulation without SDEs. Liu et al. (2023) developed I$^2$SB for image-to-image translation. For unpaired translation, Kim et al. (2024) applied neural SB, while Bortoli et al. (2024) proposed SB Flow.

**Multimodal Generation and Diffusion Models.** Diffusion models (Ho et al., 2020; Song et al., 2021) have become dominant in generative modeling, with text-to-image methods including DALL-E 2 (Ramesh et al., 2022), Latent Diffusion Models (Rombach et al., 2022), and Imagen (Saharia et al., 2022). Text-guided image editing approaches include InstructPix2Pix (Brooks et al., 2023), Prompt-to-Prompt (Hertz et al., 2022), and SDEdit (Meng et al., 2022). Cross-modal methods like BLIP-2 (Li et al., 2023) bridge vision and language, while ALAE (Pidhorskyi et al., 2020) provides the latent autoencoder we build upon.

**Knowledge Distillation.** Knowledge distillation (Hinton et al., 2015) transfers knowledge from teacher to student networks through various approaches including feature matching (Romero et al., 2015), attention transfer (Zagoruyko & Komodakis, 2017), and contrastive representation distillation (Tian et al., 2020). Recent advances include DKD (Zhao et al., 2022) which decouples target and non-target knowledge, and applications to diffusion models (Yin et al., 2024). While some works have explored connections between KD and optimal transport, our MSSB-KD provides a principled multi-marginal formulation with explicit Information Bottleneck interpretation.

# 3 BACKGROUND ON $f$-DIVERGENCE SCHRÖDINGER BRIDGE

We first recall some background on the multi-marginal Schrödinger Bridges.

**Multi-marginal $f$-divergence static Schrödinger Bridges.** The static Schrödinger bridge (SSB) is a variational problem for finding an optimal joint probability distribution, say $\pi^*$, that matches a given set of marginals while being as close as possible to a known prior distribution, say $\mathcal{R}$. We consider a general setting where this "closeness" is measured using the $f$-divergence, which quantifies the difference between two distributions. For a convex function $f$, the $f$-*divergence* of a probability measure $\mathcal{H}$ from a measure $\mathcal{R}$ on the same probability space is $D_f(\mathcal{H} \| \mathcal{R}) := \int f\left(\frac{d\mathcal{H}}{d\mathcal{R}}\right) d\mathcal{R}$ if $\mathcal{H}$ is absolutely continuous w.r.t. $\mathcal{R}$ and $\infty$ otherwise, where $\frac{d\mathcal{H}}{d\mathcal{R}}$ denotes the Radon-Nikodym derivative of $\mathcal{H}$ w.r.t. $\mathcal{R}$. For a probability space $\mathcal{X}$, let $\mathcal{P}(\mathcal{X})$ denote the set of all probability measures on $\mathcal{X}$.

Now consider $N$ probability distributions $\mu_i$ on probability spaces $\mathcal{X}_i$ for $i = 1, \ldots, N$. Denote $\Pi(\mu_1, \ldots, \mu_N)$ to be the set of joint distributions (couplings) on the product space $\mathcal{X}_1 \times \cdots \times \mathcal{X}_N$ whose $i^{th}$ marginal is $\mu_i$ for $i = 1, \ldots, N$. Let $\mathcal{R} \in \mathcal{P}(\mathcal{X}_1 \times \cdots \times \mathcal{X}_N)$ be a given prior (reference) distribution. The $f$-divergence *multi-marginal static Schrödinger bridge* (MSSB) between marginal distributions $\mu_1, \cdots, \mu_N$ w.r.t. $\mathcal{R}$ is the solution to

$$\pi^* = \arg \min_\pi D_f(\pi \| \mathcal{R}) \quad \text{subject to} \quad \pi \in \Pi(\mu_1, \ldots, \mu_N). \tag{1}$$

It becomes the standard KL-divergence SSB when $f(x) = x \log x$. In this case, equation 1 becomes the classical SSB problem (see, e.g., Fortet (1940); Pavon et al. (2021)). SSBs with general $f$-divergence have been studied more recently Carlier et al. (2017); Lorenz & Mahler (2022); Terjék & González-Sánchez (2022). When

$$d\mathcal{R}(x_1, \ldots, x_N) \propto \exp(-c(x_1, \ldots, x_N)/\epsilon) \, d\mu_1(x_1) \otimes \cdots \otimes \mu_N(x_N) \tag{2}$$

for some cost function $c$ and a regularization parameter $\epsilon > 0$, this becomes the multi-marginal version of the celebrated entropic optimal transport (EOT) problem Villani (2021).

**Kantorovich dual.** Let $\psi(y) = \sup_x (yx - f(x))$ denote the convex conjugate of $f$. It is well-known that the $f$-divergence SSB admits the following Kantorovich dual

$$\sup_{\alpha_1, \ldots, \alpha_N} \left( \sum_{i=1}^N \int \alpha_i(x_i) \, d\mu_i(x_i) \right) - \int \psi(\oplus_{i=1}^N \alpha_i(\mathbf{x})) \, d\mathcal{R}(\mathbf{x}), \tag{3}$$

where $\mathbf{x} = (x_1, \ldots, x_N)$ and $\oplus_{i=1}^N \alpha_i(\mathbf{x}) = \sum_{i=1}^N \alpha_i(x_i)$. This is a concave maximization problem in the dual variables (potentials) $\alpha_1, \ldots, \alpha_N$. When $N = 2$, alternating maximization Bertsekas (1997); Beck & Tetruashvili (2013a) for this dual problem for KL-divergence (i.e., $\psi(\cdot) = \exp(\cdot)$) is precisely the celebrated Sinkhorn algorithm. Once the potentials $\alpha_1, \ldots, \alpha_N$ are found, the MSSB $\pi^*$ is recovered through the relation

$$d\pi^*(\mathbf{x}) = \psi'(\oplus_{i=1}^N \alpha_i(\mathbf{x})) \, d\mathcal{R}(\mathbf{x}). \tag{4}$$

In particular, with KL-divergence and Gibbs prior equation 2, $\psi(\cdot) = \exp(\cdot)$ so equation 4 becomes the familar EOT form

$$d\pi^*(\mathbf{x}) = \exp(\oplus_{i=1}^N \alpha_i(\mathbf{x}) - c(\mathbf{x})/\epsilon) d(\mu_1 \otimes \cdots \otimes \mu_N)(\mathbf{x}). \tag{5}$$

**Dynamic Schrödinger bridges.** The classic *dynamic Schrödinger bridge* (DSB) problem is to learn a stochastic process $T^*$ on a domain $\Omega$ such that its distributions at times $t = 0, 1$ match the given marginals $\mu_0, \mu_1$ and is as close as possible to a prior process, often taken to be the Wiener process

$W^\epsilon$ with volatility $\epsilon > 0$. The DSB $T^*$ with the Wiener prior $W^\epsilon$ between marginals $\mu_0, \mu_1$ is the minimizer of the following variational problem: taking the KL-divergence with $f(x) = x \log x$,

$$T^* = \arg\min_T D_f(T \parallel W^\varepsilon) \quad \text{subject to} \quad T \in \mathcal{F}(\mu_0, \mu_1), \tag{6}$$

where $\mathcal{F}(\mu_0, \mu_1)$ is the set of stochastic processes on $\Omega$ which start at distribution $\mu_0$ (at $t = 0$) and end at $\mu_1$ (at $t = 1$). The essence of a DSB lies in the corresponding SSB obtained by forgetting all intermediate times $0 < t < 1$, which is equivalent to the EOT with the quadratic loss $c(x, y) = \|x - y\|^2$. Conversely, one can construct the full DSB by sampling the initial and target locations $(x, y)$ from the SSB $\pi^*$ and connecting them by the (information-less) Brownian bridge. However, this equivalence between DSB and SSB is not known to hold for general $f$-divergences. We prove that this equivalence is indeed true for general $f$-divergences.

For the $N = 2$ case, DSB has been more popular than SSB for some computational benefits. Namely, DSB can leverage the reciprocal and Markov properties of the prior process, which are highly effective for developing computational algorithms. As a result, most of the computational methods in the literature are based on the dynamic formulation Bortoli et al. (2023); Liu et al. (2023); Chen et al. (2021); Vargas et al. (2021). However, it is not clear how to formulate multi-marginal version of DSB, which was straightforward for the SSB. In fact, we will argue that working directly with MSSB can be extremely flexible and computationally efficient.

## 4 ESTIMATING MSSB VIA KANTOROVICH DUAL AND BLOCK OPTIMIZATION

We first propose a general method for estimating MSSB from empirical data, which we will use for training specific instances of our model in later sections.

Since in practical applications one has to work with finite samples, it is reasonable to assume that we only have access to empirical marginal distributions, say $\hat\mu_i$, $i = 1, \ldots, N$, and do not know the population marginals $\mu_i$, but we still seek to estimate the population MSSB $\pi^*$ via some approximate coupling $\hat\pi$ defined on the whole space $\mathcal{X}_1 \times \cdots \times \mathcal{X}_N$. According to the Kantorovich duality, it is enough to estimate the population potentials $\alpha_1, \ldots, \alpha_N$ as $\hat\alpha_1, \ldots, \hat\alpha_N$, since then we can set $d\hat\pi = \psi'(\hat\alpha_1 \oplus \cdots \oplus \hat\alpha_N)\, d\mathcal{R}$. One often parametrizes some transform of the potentials and fits it with the empirical data (e.g., Gaussian mixture Korotin et al. (2023)). Our final MSSB solution $\hat\pi$ allows out-of-sample evaluation, which is critical for inference tasks in generative modeling.

Even though the multi-marginal Sinkhorn algorithm is known to exhibit linear convergence for KL-divergence Carlier (2022), computing the Sinkhorn updates can be intractable for continuous or very high dimensional potentials. Due to the multi-modality, we cannot use the DSB formulation as usually done in the literature for $N = 2$ case Bortoli et al. (2023); Liu et al. (2023) . Even in that case, existing approaches base heavily on the structure of EOT with KL-divergence and quadratic cost $c(x, y) = \|x - y\|^2$ (e.g., Korotin et al. (2023; 2024)), but we potentially use nonconvex task-specific costs as in equation 9.

Instead, we propose to parameterize the exponentiated potentials in 5, $\exp(\alpha_i(x)) \approx u_{\theta_i}(x)$ by a suitable neural network $u_{\theta_i}$ with parameters $\theta_i$[1]. These parameters are optimized by minimizing the Kantorovich dual loss on the empirical data:

$$\sup_{\theta_1, \ldots, \theta_N} \underbrace{\left( \sum_{i=1}^N \int \log u_{\theta_i}(x_i)\, d\hat\mu_i(x_i) \right) - \int \psi(\oplus_{i=1}^N \log u_{\theta_i}(\mathbf{x}))\, d\hat{\mathcal{R}}(\mathbf{x})}_{\mathcal{L}(\theta)}. \tag{7}$$

While the above problem could be highly non-concave, we leverage its block structure and propose to use cyclic block optimization (e.g., Lyu & Li (2025); Wright (2015)) for simple and efficient optimization. For instance, at each training step, we may sample a minibatch of data from the product of the empirical marginals, evaluate the empirical loss and its gradients, and update the neural net parameters $\theta_i$ accordingly. This optimization procedure (for $N = 3$) is summarized in Algorithm 1 (provided in Appendix A.3) and can easily be implemented using GPUs in parallel.

---

[1]We model each exponentiated potential by a multi-layer perceptron (MLP).

## 5 STYLIZED MULTI-MARGINAL STATIC SCHRÖDINGER BRIDGES.

### 5.1 MULTI-MODAL GENERATIVE MODELING

We extend the Static Schrödinger Bridge (SSB) framework to handle multiple modalities, enabling conditional generation across heterogeneous sources such as image and text. Our framework generates samples that **(i)** incorporate information from multiple modalities, and **(ii)** remain faithful to the content or style of the source. Formally, we treat each modality as a marginal distribution $(\mu_1, \mu_2, \dots)$ within a globally normalized probabilistic model. We then sample the target modality via the conditional distribution given the different heterogeneous source distributions. This multi-marginal formulation allows us to encode cross-modal relationships as entropic optimal transport couplings, laying the foundation for our generative objective.

**Formulation.** For a concrete discussion and clarity of presentation, consider the following specific application context with $N = 3$ MSSB for text-guided image generation. We consider a setting where the source and target images are sampled from distinct but related distributions (e.g., different domains, styles, or identities), and the textual prompt defines the transformation semantics. This enables a wide range of tasks including style-preserving edits, identity-conditioned transfers, and semantic image translation. We seek to learn the population 3-mode MSSB $\pi^*$ over distributions of (latent) input images $\mu_1 \in \mathcal{P}(\mathcal{X}_{\text{input}})$, (latent) output images $\mu_2 \in \mathcal{P}(\mathcal{Y}_{\text{output}})$, and (latent) text instruction $\mu_3 \in \mathcal{P}(\mathcal{T}_{\text{instr}})$ with the prior measure $\mathcal{R}$ defined in the following Gibbs form with cost function $c(\cdot, \cdot, \cdot)$ and regularization parameter $\epsilon$:

$$d\mathcal{R}(x, y, t) \propto \exp\left(-c(x, y, t)/\epsilon\right) \, d\mu_1(x) \, d\mu_2(y) \, d\mu_3(t). \tag{8}$$

The core design of our framework lies in defining a suitable (potentially highly nonconvex) cost function in equation 8. A specific form that satisfies our requirements is given by

$$c(x, y, t) = \lambda_1 \|x - y\|_2^2 + \lambda_2 \|y - M_\psi(t)\|_2^2 + \lambda_3 \|y\|_2^2, \tag{9}$$

where $\|\cdot\|_2$ denotes the standard $\ell_2$ norm and $\lambda_1$, $\lambda_2$ and $\lambda_3$ denote the tradeoff parameters (we may set $\lambda_1 = 1$). This cost balances fidelity to the source (latent) image $x$ with semantic alignment to the text prompt $t$, which is mapped into an (latent) image space via a pre-trained model $M_\psi(t)$, such as CLIP2GAN Wang et al. (2023) (see Sec. 7.1 for more details). The third term acts as a regularizer to keep the generated latent within bounded norm. Our cost function encodes semantic alignment and structural fidelity across modalities, while the temperature parameter $\epsilon$ balances transport fidelity with diversity.

**Inference.** At test time, our goal is to sample from the conditional multi-marginal distribution $\pi^*(y \mid x^*, t^*)$ defined by the optimal Kantorovich potentials. This distribution takes the form of a Gibbs measure over image latents:

$$d\pi^*(y \mid x^*, t^*) \propto \exp(\beta(y) - c(x^*, y, t^*)/\epsilon) \, d\mu_2(y),$$

where $\beta(y)$ is the learned dual potential, $c(x, y, t)$ is the semantic cost function, and $\mu_2$ is the base measure over latent space. Since we may not know $\mu_2$ above, we may replace it by some other priors (e.g., uniform distribution).

The resulting density is typically high-dimensional, non-Gaussian, and lacks a tractable normalization constant, making direct sampling intractable. To address this, we adopt *Langevin dynamics*, a gradient-based Markov Chain Monte Carlo (MCMC) technique that naturally exploits the structure of the log-unnormalized Gibbs distribution. By iteratively updating $y$ via noisy gradient ascent on the log-density, Langevin dynamics enables effective sampling from $\pi^*(y \mid x^*, t^*)$ while preserving the semantic alignment and fidelity properties induced by the Schrödinger Bridge formulation. Empirically, we find that Unadjusted Langevin Algorithm (ULA) enables stable and diverse sampling, allowing the generated latents to faithfully reflect the semantic intent of the input text while preserving alignment with source identity and structural constraints. The procedure is summarized in Algorithm 2 in Appendix A.3.

### 5.2 KNOWLEDGE DISTILLATION

**Formulation.** Next, we showcase another compelling application of our MSSB framework in knowledge distillation (KD). Here, the goal is to transfer information from one or more teacher

networks to a student network. Classical KD methods typically minimize a combination of cross-entropy and KL divergence between teacher and student outputs, but they lack a unifying probabilistic interpretation. We show that MSSB provides such an interpretation by treating data, teacher logits, and student logits as three marginals in the SB problem with a Gibbs prior measure - equivalent to an EOT problem.

Let $\mathcal{X}$ denote the data space, $\mathcal{T}$ the teacher logit space, and $\mathcal{S}$ the student logit space with marginals $\mu_{\mathcal{X}}, \mu_{\mathcal{T}}$, and $\mu_{\mathcal{S}}$, respectively. The MSSB-KD objective seeks:

$$\min_{\pi \in \Pi(\mu_{\mathcal{X}}, \mu_{\mathcal{T}}, \mu_{\mathcal{S}})} \mathrm{KL}(\pi \| R) = \min_{\pi} \left\{ \int c(x,t,s)d\pi + \epsilon \cdot D_{\mathrm{KL}}(\pi \| \mu_{\mathcal{X}} \otimes \mu_{\mathcal{T}} \otimes \mu_{\mathcal{S}}) \right\} \quad (10)$$

where we define $c(x,t,s) = \|s-t\|_2^2 + \lambda_{\mathrm{ce}}\mathcal{L}_{\mathrm{CE}}(s,t)$ to measure the discrepancy between the teacher and student logits, and $dR(x,t,s) \propto \exp\left(-\frac{c(x,t,s)}{\epsilon}\right) d\mu_{\mathcal{X}}(x)\,d\mu_{\mathcal{T}}(t)\,d\mu_{\mathcal{S}}(s)$. Here, $\mathcal{L}_{\mathrm{CE}}(s,t)$ is the cross-entropy loss and $\lambda_{ce}$ is the corresponding weight parameter.

This objective is exactly the MSSB functional specialized to KD: it balances fidelity to teacher outputs (via the cost term) with compression of the student representation (via entropy regularization). Our knowledge distillation approach, MSSB-KD, matches the test accuracies of standard baseline models (see Table 1-Left), is robust to noisy teacher, and beats standard KD baselines on multi-teacher distillation (see Table 1-Right). We further show that our MSSB-KD can be interpreted as minimizing a variational relaxation of the Information Bottleneck objective (see Theorem 2).

## 6 THEORETICAL RESULTS

**Equivalence of Dynamic and Static Schrödinger Bridges**: We prove a key theoretical improvement in the theory of Dynamic Schrödinger Bridges (DSB) and Static Schrödinger Bridges (SSB) that arises the possibility of choosing the most suitable $f$-divergence for a given task or, perhaps, even parameterizing the divergence itself. The proof establishes that the infimum of the $f$-divergence DSB over the space of path measures is equal to the infimum of the $f$-divergence SSB over the space of endpoint joint measures. For simplicity, we work with the two marginal case but the proof extends to multiple marginals. Our key result is provided below; we defer its proof to Appendix A.1.1.

**Theorem 1** (Equivalence of DSB and SSB). *The infimal values of the dynamic and static $f$-divergence minimization problems are equal:*

$$\inf_{\Pi \in \mathcal{C}(\mu_0, \mu_1)} D_f(\Pi \| P) = \inf_{\pi \in \Pi(\mu_0, \mu_1)} D_f(\pi \| P_{01})$$

*where $\mathcal{C}(\mu_0, \mu_1)$ is the set of all path measures on $\Omega$ whose endpoint distributions belong to $\Pi(\mu_0, \mu_1)$.*

**MSSB-KD as Variational Information Bottleneck**: We now establish the connection between MSSB-based Knowledge Distillation (MSSB-KD) and the Information Bottleneck principle. More specifically, while not exactly equivalent to the classic Information Bottleneck formulation $\min_{p(s|x)}[I(S;X) - \lambda I(S;T)]$, MSSB-KD achieves the same goal through optimal transport: finding a student representation that compresses input information while preserving teacher knowledge. The total correlation term $TC(X;T;S)$ measures the total correlation in $\pi$, penalizing unnecessary dependencies. We now state our result - proof and further discussion are deferred to Appendix A.2.

**Theorem 2.** *The MSSB-KD objective implements a variational relaxation of the Information Bottleneck principle. Specifically, minimizing MSSB-KD is equivalent to minimizing*

$$J_{MSSB\text{-}KD}(\pi) = \underbrace{\mathbb{E}_{\pi}[\|s-t\|^2 + \lambda_{ce}\mathcal{L}_{CE}(s,t)]}_{\text{Distortion penalty}} + \epsilon \cdot \underbrace{TC(X;T;S)}_{\text{Total correlation}} + \epsilon \cdot \underbrace{\sum_{i \in \{X,T,S\}} KL(\pi_i \| \mu_i)}_{\text{Marginal regularization}}. \quad (11)$$

## 7 EXPERIMENTS

We evaluate the effectiveness of our proposed MSSB framework on two fundamental tasks in multimodal generation: (1) **text-to-image generation** from pure noise, where the goal is to synthesize images that semantically align with a given text prompt; and (2) **text-guided image editing**, where

the task is to modify a source image based on textual instructions while preserving its underlying content or style. Our experiments aim to demonstrate the flexibility, expressiveness, and sample efficiency of MSSB. MSSB is capable of generating diverse and semantically aligned outputs, with explicit control over image-text faithfulness ($\lambda_2$) and sample diversity ($\epsilon$), highlighting its adaptability to different generative goals.

## 7.1 EXPERIMENTAL SETUP

We use the FFHQ dataset Karras et al. (2019) as the image source. For text supervision, we leverage the `ffhq512-caption` dataset SJTU (2023) from HuggingFace, which provides human-written captions for each FFHQ image. These captions include descriptions such as facial attributes, age, and accessories (e.g., "a smiling man with short hair", "a woman with bangs and sunglasses"). Each image-text pair is used to construct a triplet $(x, y, t)$, where $x$ and $y$ are source and target image latents, and $t$ is the guiding text embedding. We use the pretrained ALAE autoencoder Pidhorskyi et al. (2020) to extract the 512-dimensional latent representation of each image. This latent serves as the domain where we solve the Schrödinger Bridge for image generation and editing.

**Mapper Network** To effectively align textual instructions with image representations in latent space, we introduce a cross-attention-based mapper that plays a central role in our framework. As illustrated in Figure 3 (in Appendix A.5), the mapper serves as a bridge from the CLIP image-text embedding space to the ALAE latent space, enabling our cost function to integrate semantic guidance directly from text. The input token embeddings, $E_t \in \mathbb{R}^{T \times d}$, are obtained from a frozen text encoder $\phi_T(\cdot)$, instantiated as CLIP's ViT-B/32 text encoder (Ahn et al. (2019)). Unlike previous approaches such as Wang et al. (2022), which rely solely on a feed-forward network (MLP) to map from CLIP embeddings to generative latents, our mapper incorporates a stack of Transformer-style blocksVaswani et al. (2017) with *multi-head cross-attention*. This design allows the mapper to directly condition on token-level semantics, rather than collapsing the entire text sequence into a single embedding prematurely.

## 7.2 TEXT-TO-IMAGE GENERATION FROM NOISE

To evaluate the ability of our MSSB framework to perform unconditional generation guided purely by text, we conduct experiments where the source input is a pure Gaussian noise image $z \in \mathbb{R}^{1024 \times 1024 \times 3}$, with each pixel independently sampled from a standard normal distribution. This noise image is encoded by the pretrained ALAE encoder to obtain the initial latent representation. At inference time, given a user-specified caption $t$ (e.g., "an old man with glasses"), we compute the target latent $y$ by solving the Schrödinger Bridge between $x$ and $y$ under the guidance of $t$. The target $y$ is then decoded via the ALAE decoder to generate the final image. This setup enables the synthesis of high-quality human faces that align well with arbitrary text descriptions, without requiring an input image. As shown in Figure 1, our MSSB method produces diverse and faithful samples, demonstrating strong semantic controllability and multimodal reasoning. For completeness, we include additional diverse generations in Appendix A.7.

## 7.3 TEXT-GUIDED IMAGE EDITING

In addition to generating faces from noise, our MSSB framework also supports fine-grained text-guided editing of existing images. In this setting, the source image latent $x$ is obtained by encoding a real image using the pretrained ALAE encoder, and the text $t$ describes the desired modification (e.g., "make the person smile" or "add glasses"). We then solve the MSSB to compute a new latent $y$ that minimizes the cost $c(x, y, t)$, thereby producing an edited version of the input image that aligns with the textual instruction while preserving identity and style from the original. The resulting $y$ is decoded into the final image using the ALAE decoder. This setup offers a flexible and semantically controllable approach to image editing without explicit attribute supervision. As illustrated in Figure 2, our method can perform meaningful edits such as changing expression, age, or accessories, while preserving key facial features. Furthermore, by adjusting $\lambda_2$ and $\varepsilon$, users can control the trade-off between the degree of semantic editing and fidelity to the original image, enabling both subtle and substantial modifications under the same framework.

A smiling woman with curly hair, hoop earrings, red lipstick

An elderly man with gray hair, round glasses, serious look

A young boy with short black hair, cheerful expression

A young man with short brown hair, pointed chin, smiling

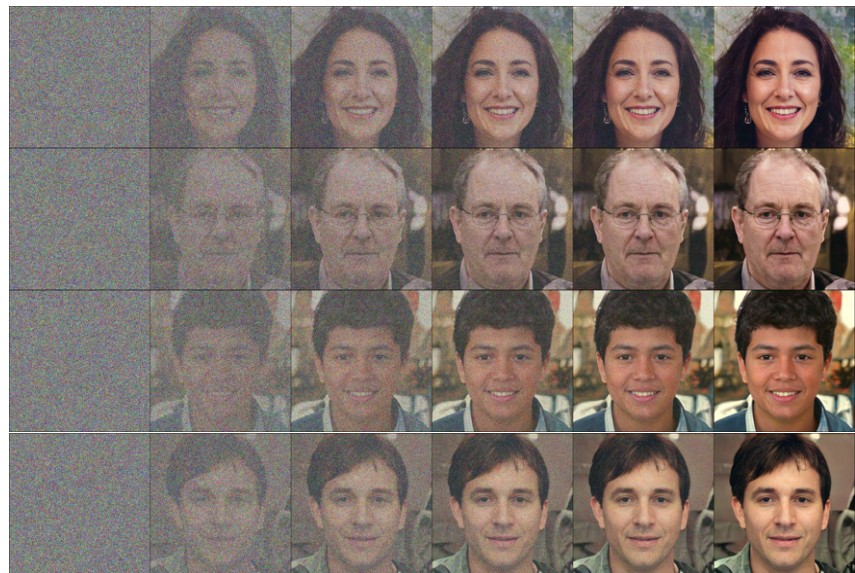

Figure 1: Text-to-image synthesis results using our MSSB framework. Given only textual descriptions (left), our method generates semantically faithful and high-quality human face images (right). The intermediate visualizations are obtained by Brownian Bridge sampling, illustrating a smooth semantic transition during generation.

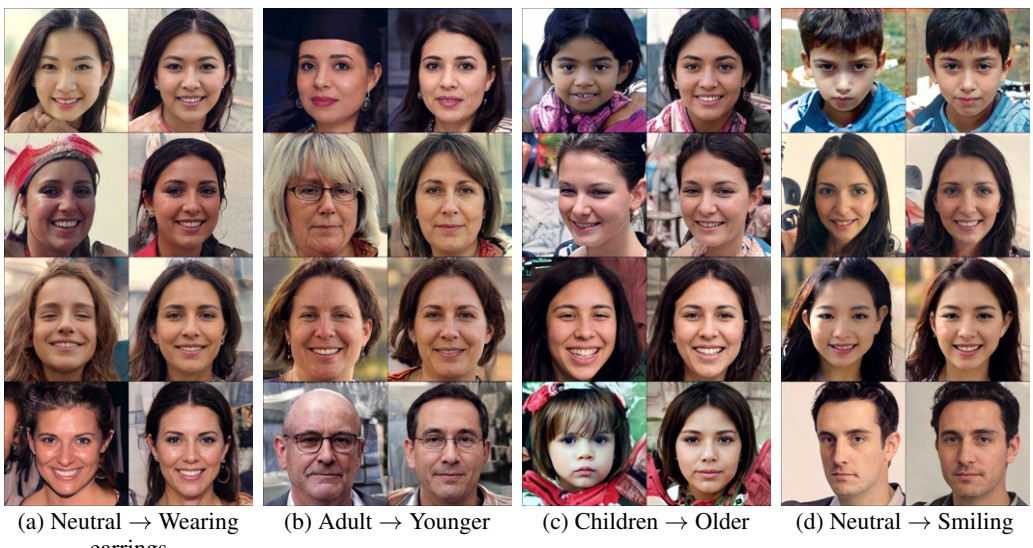

(a) Neutral → Wearing earrings  (b) Adult → Younger  (c) Children → Older  (d) Neutral → Smiling

Figure 2: **Text-guided image editing.** Each panel shows several "original (left) → edited (right)" pairs. Prompts are *wearing earrings*, *younger*, *older*, and *smiling*.

## 7.4 KNOWLEDGE DISTILLATION WITH MSSB-KD

We evaluate MSSB-KD on standard knowledge distillation benchmarks to demonstrate its effectiveness in knowledge transfer and its unique advantages over conventional approaches. In our experiments, we use standard teacher-student pairs where WRN denotes Wide ResNet architectures The goal is to transfer knowledge from these larger, more accurate teachers to smaller, deployable students.

**Implementation Details.** We solve the three-marginal optimal transport between data $x$, teacher logits $t$, and student logits $s$ with cost $c(x, t, s) = \|s - t\|^2 + \lambda_{ce}\mathcal{L}_{CE}(s, t)$. The entropy parameter $\epsilon$

follows a cosine schedule from 1.0 to 0.1, implementing our theoretical insight that high initial entropy encourages exploration before refinement. This adaptive scheduling proved crucial for balancing the compression-fidelity trade-off inherent in the Information Bottleneck formulation. Table 1 compares MSSB-KD against KD (Hinton et al., 2015), AT (Zagoruyko & Komodakis, 2017), CRD (Tian et al., 2020), and DKD (Zhao et al., 2022).

| Method | WRN-40-2 → WRN-16-2 | | ResNet-56 → ResNet-20 | | ResNet-110 → ResNet-32 | |
|---|---|---|---|---|---|---|
| | T-1 | T-5 | T-1 | T-5 | T-1 | T-5 |
| Teacher | 75.6 | 93.0 | 72.3 | 91.4 | 74.3 | 92.0 |
| Student | 68.4 | 88.8 | 69.1 | 88.9 | 71.1 | 90.6 |
| KD | 70.8 | 89.9 | 70.7 | 89.9 | 73.1 | 91.6 |
| AT | 70.6 | 90.0 | 70.2 | 89.6 | 72.8 | 91.4 |
| CRD | 71.2 | 90.1 | 71.4 | 90.4 | 73.5 | 91.8 |
| DKD | **72.0** | **90.4** | **71.8** | **90.6** | **74.1** | **92.0** |
| MSSB | 71.7 | 90.3 | 71.5 | 90.4 | 73.9 | 91.9 |

**(a) Noisy Teacher (WRN-40-2 → WRN-16-2)**

| Method | Clean | $\sigma$=0.1 | $\sigma$=0.3 | $\sigma$=0.5 | $\sigma$=1.0 |
|---|---|---|---|---|---|
| KD | 70.8 | 69.3 | 65.9 | 61.0 | 52.4 |
| CRD | 71.2 | 69.9 | 66.8 | 62.4 | 54.1 |
| DKD | 72.0 | 70.5 | 67.9 | 63.8 | 55.9 |
| MSSB | **71.7** | **70.9** | **68.4** | **65.2** | **58.7** |

**(b) Multi-Teacher (3 teachers → ResNet-32)**

| Method | Aggr. | CIFAR-100 | CIFAR-10 |
|---|---|---|---|
| Best | - | 73.5 | 93.4 |
| KD | Avg | 73.8 | 93.7 |
| KD | Wtd | 74.0 | 93.8 |
| MSSB | OT | **74.9** | **94.3** |

Table 1: Knowledge distillation results on CIFAR-100. **Left**: Test accuracy comparison across different teacher-student pairs. **Right**: (a) Robustness to noisy teachers with varying noise levels $\sigma$. (b) Multi-teacher distillation performance. MSSB-KD shows superior robustness and effective multi-teacher knowledge aggregation.

**Robustness Analysis.** A critical advantage of MSSB-KD emerges in challenging scenarios. The right panel (Table 1(a)) evaluates robustness to noisy teacher supervision, where $\sigma$ represents the standard deviation of Gaussian noise added to teacher logits. This simulates real-world scenarios where teacher models may be uncertain or partially corrupted. The entropy regularization in MSSB-KD acts as a natural denoising mechanism: while standard KD degrades catastrophically (dropping to 52.4% at $\sigma = 1.0$), MSSB-KD maintains 58.7% accuracy.

**Multi-Teacher Distillation.** Table 1(b) showcases MSSB-KD's unique capability to handle $M$ teachers through $(M + 2)$-marginal MSSB-KD where the two additional marginals are for the data and the student logits. While standard KD requires ad-hoc aggregation strategies (averaging or weighted combinations), MSSB-KD naturally finds the optimal transport plan across all teachers simultaneously.

**Key Insights.** Our experiments reveal three fundamental advantages of MSSB-KD: (1) The entropy regularization provides implicit noise robustness without additional engineering, (2) The multi-marginal formulation elegantly handles multiple teachers without heuristic aggregation, and (3) The Information Bottleneck connection ensures optimal compression-fidelity trade-offs. These results validate our theoretical framework and demonstrate that viewing knowledge distillation through the lens of MSSB opens new avenues for robust and efficient model compression.

## 8 CONCLUSION

We introduce Multi-marginal Static Schrödinger Bridges (MSSB) as a unified framework for multi-modal generation and knowledge distillation, extending beyond two-marginal constraints to handle three or more distributions simultaneously. Our theoretical contributions include proving equivalence between static and dynamic formulations for general $f$-divergences and establishing that MSSB-KD implements Information Bottleneck principles. Experiments demonstrate MSSB's effectiveness for text-guided image generation with controllable diversity through entropy regularization, and superior robustness in knowledge distillation scenarios including noisy teachers and multi-teacher settings. The framework's flexibility in cost function design and divergence selection opens promising avenues for task-specific static Schrödinger Bridge/Optimal Transport solutions across diverse machine learning applications.

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

# A APPENDIX

## A.1 EQUIVALENCE OF DYNAMIC AND STATIC SCHRÖDINGER BRIDGES

We provide a rigorous proof for the equivalence of the dynamic and static formulations of the Schrödinger Bridge problem for a general $f$-divergence. The proof establishes that the infimum of the $f$-divergence over the space of path measures is equal to the infimum over the space of endpoint joint measures. Before we dive into the proof in section A.1.1, we state a few key definitions and preliminaries.

**Definition 1** (Spaces and Measures). *Let $(\Omega, \mathcal{F})$ be a measurable space of paths, and let $(\mathcal{X}, \mathcal{B})$ be the state space at a single point in time. We assume $\Omega$ and $\mathcal{X} \times \mathcal{X}$ are Polish spaces equipped with their Borel $\sigma$-fields.*

- ***Path Measures:** Let $P \in \mathcal{P}(\Omega)$ be a prior probability measure on the path space (e.g., Wiener measure). Let $\Pi \in \mathcal{P}(\Omega)$ be a candidate path measure that is absolutely continuous with respect to $P$ ($\Pi \ll P$).*

- ***Projection Map:** Let $\phi : \Omega \to \mathcal{X} \times \mathcal{X}$ be the measurable map that extracts the endpoints of a path $\omega \in \Omega$, such that $\phi(\omega) = (\omega_0, \omega_1)$.*

- ***Endpoint Measures:** The projection map $\phi$ induces pushforward measures on the product space $\mathcal{X} \times \mathcal{X}$. Let $P_{01} := \phi_\# P$ and $\pi := \phi_\# \Pi$ be the respective endpoint (or static) measures.*

**Definition 2** ($f$-Divergence). *Let $f : \mathbb{R}^+ \to \mathbb{R}$ be a convex function with $f(1) = 0$. The **f-divergence** of a measure $\Pi$ from a measure $P$ (where $\Pi \ll P$) is*

$$D_f(\Pi \| P) := \int_\Omega f\left(\frac{d\Pi}{dP}(\omega)\right) dP(\omega) = \mathbb{E}_P\left[f\left(\frac{d\Pi}{dP}\right)\right]$$

*The Kullback-Leibler (KL) divergence is a special case with $f(t) = t \log t$.*

**Definition 3** (Static and Dynamic Schrödinger Bridges). *Given fixed marginal distributions $\mu_0, \mu_1 \in \mathcal{P}(\mathcal{X})$, we define the constraint sets:*

- *$\Pi(\mu_0, \mu_1)$: The set of all joint measures (couplings) on $\mathcal{X} \times \mathcal{X}$ with marginals $\mu_0$ and $\mu_1$.*

- *$\mathcal{C}(\mu_0, \mu_1)$: The set of all path measures $\Pi$ on $\Omega$ whose endpoint distribution $\phi_\# \Pi$ belongs to $\Pi(\mu_0, \mu_1)$.*

*The optimization problems are:*

$$\textbf{Static Problem (SSB):} \quad \inf_{\pi \in \Pi(\mu_0, \mu_1)} D_f(\pi \| P_{01}) \tag{12}$$

$$\textbf{Dynamic Problem (DSB):} \quad \inf_{\Pi \in \mathcal{C}(\mu_0, \mu_1)} D_f(\Pi \| P) \tag{13}$$

**Proposition 3** (Disintegration Formula, Léonard (2013a)). *For any measurable function $\phi : \Omega \to \mathcal{X} \times \mathcal{X}$ and any measure $P \in \mathcal{P}(\Omega)$, the disintegration formula is defined as:*

$$dP(\omega) = dP(\cdot|x, y)dP_{01}(x, y) \tag{14}$$

**Proposition 4** (Léonard (2013a), Theorem 1.6). *$\mathbb{E}_P(\cdot|\phi)$ admits a regular conditional probability kernel $x \in \mathcal{X} \times \mathcal{X} \to P(\cdot|\phi = (x, y)) \in \mathcal{P}(\Omega)$.*

**Proposition 5** (Léonard (2013a), Proposition A.13). *Let $\phi_\# \Pi = \Pi_\phi$, $\phi_\# P = P_\phi$. Suppose $P_\phi \in \mathcal{P}(\mathcal{X})$ is $\sigma$-finite. Suppose $P$ is bounded and $\Pi \ll P$, then $\Pi_\phi \ll P_\phi$ and*

$$\frac{d\Pi_\phi}{dP_\phi}(\phi) = \mathbb{E}_P\left(\frac{d\Pi}{dP}|\phi\right), \quad P\text{-a.e.} \tag{15}$$

*and for any bounded measurable function $f$:*

$$\mathbb{E}_\Pi(f|\phi)\mathbb{E}_P\left(\frac{d\Pi}{dP}|\phi\right) = \mathbb{E}_P\left(\frac{d\Pi}{dP}f|\phi\right), \quad P\text{-a.e.} \tag{16}$$

**Theorem 6** (Léonard (2013a), Theorem 1.6). *Suppose $\Pi \ll P$, and $P_\phi$ and $\Pi_\phi$ are $\sigma$-finite measures on $\mathcal{X} \times \mathcal{X}$, then $\Pi(\cdot|\phi) \ll P(\cdot|\phi)$, $\Pi$-a.e., and*

$$\frac{d\Pi}{dP}(\omega) = \frac{d\Pi(\cdot|\phi = \phi(\omega))}{dP(\cdot|\phi = \phi(\omega))}(\omega)\frac{d\Pi_\phi}{dP_\phi}(\phi(\omega)) \qquad \forall \omega \in \Omega, \Pi - a.e. \tag{17}$$

*or, more specifically, for our choice of projection map $\phi$ that extracts the endpoints of a path $\omega \in \Omega$ we have*

$$\frac{d\Pi}{dP}(\omega) = \frac{d\Pi(\cdot|\phi(\omega))}{dP(\cdot|\phi(\omega))}(\omega)\frac{d\Pi_{01}}{dP_{01}}(\phi(\omega)) \tag{18}$$

Throughout, $\Omega$ and $\mathcal{X} \times \mathcal{X}$ are Polish with Borel $\sigma$-fields; regular conditional probabilities exist; and whenever $D_f(\pi\|P_{01}) < \infty$ we have $\pi \ll P_{01}$. We do not require $P$ to be bounded; absolute continuity and finiteness of $D_f$ ensure the needed integrability.

### A.1.1 PROOF OF THEOREM 1

We now restate Theorem 1 and present its proof.

**Theorem 7** (Theorem 1, Equivalence of DSB and SSB). *The infimal values of the dynamic and static f-divergence minimization problems are equal:*

$$\inf_{\Pi \in \mathcal{C}(\mu_0, \mu_1)} D_f(\Pi\|P) = \inf_{\pi \in \Pi(\mu_0, \mu_1)} D_f(\pi\|P_{01})$$

*Proof.* Let us denote the RHS of the equality by $\inf_{\text{static}}$ and the LHS by $\inf_{\text{dynamic}}$, respectively. We begin by showing the inequality $\inf_{\text{static}} \leq \inf_{\text{dynamic}}$.

**Part 1:** $\inf_{\text{static}} \leq \inf_{\text{dynamic}}$.

Let $\Pi_{01} = \phi_\#\Pi$ and $P_{01} = \phi_\#P$. By Proposition 5, the relative pushforward measure is the conditional expectation:

$$\frac{d\Pi_{01}}{dP_{01}}(x, y) = \mathbb{E}_P\left[\frac{d\Pi}{dP}\bigg|\phi(\omega) = (x, y)\right].$$

By Definition 2, the divergence of the static measures is

$$D_f(\Pi_{01}\|P_{01}) = \int_{\mathcal{X}\times\mathcal{X}} f\left(\mathbb{E}_P\left[\frac{d\Pi}{dP}\bigg|\phi = (x, y)\right]\right) dP_{01}(x, y)$$

By Jensen's inequality for convex $f$:

$$f\left(\mathbb{E}_P\left[\frac{d\Pi}{dP}\bigg|\phi\right]\right) \leq \mathbb{E}_P\left[f\left(\frac{d\Pi}{dP}\right)\bigg|\phi\right]$$

Integrating with respect to $P_{01}$ and using the law of total expectation:

$$D_f(\Pi_{01}\|P_{01}) \leq \int_{\mathcal{X}\times\mathcal{X}} \mathbb{E}_P\left[f\left(\frac{d\Pi}{dP}\right)\bigg|\phi = (x, y)\right] dP_{01}(x, y) \tag{19}$$

$$= \mathbb{E}_P\left[f\left(\frac{d\Pi}{dP}\right)\right] = D_f(\Pi\|P) \tag{20}$$

For any $\Pi \in \mathcal{C}(\mu_0, \mu_1)$, its pushforward $\pi = \phi_\#\Pi$ is in $\Pi(\mu_0, \mu_1)$. Therefore:

$$\inf_{\Pi \in \mathcal{C}(\mu_0, \mu_1)} D_f(\Pi\|P) \geq \inf_{\pi \in \Pi(\mu_0, \mu_1)} D_f(\pi\|P_{01}) \tag{21}$$

We now show the reverse inequality.

**Part 2:** $\inf_{\text{dynamic}} \leq \inf_{\text{static}}$.

Note that for any $\pi \in \Pi(\mu_0, \mu_1)$ such that $\pi \ll P_{01}$, we can define the lifted path measure

$$\Pi^\pi(A) := \int_{\mathcal{X} \times \mathcal{X}} P(A|x,y) d\pi(x,y) \qquad \forall A \in \mathcal{F} \tag{22}$$

where the regular conditional probability kernel $P(\cdot|x,y)$ is due to Proposition 4.

Then $\phi_\# \Pi^\pi = \pi$ by construction and $\Pi^\pi \ll P$. Using Proposition 3 and Theorem 6, we have

$$\frac{d\Pi^\pi}{dP}(\omega) = \frac{d\pi}{dP_{01}}(\phi(\omega)). \tag{23}$$

Therefore, we can write

$$D_f(\Pi^\pi \| P) = \int_\Omega f\left(\frac{d\Pi^\pi}{dP}\right) dP \tag{24}$$

$$= \int_{\mathcal{X} \times \mathcal{X}} f\left(\frac{d\pi}{dP_{01}}\right) dP_{01} \tag{25}$$

$$= D_f(\pi \| P_{01}) \tag{26}$$

This shows that for every static $\pi$, $\exists\, \Pi^\pi \in \mathcal{C}(\mu_0, \mu_1)$ with equal $f$-divergence which gives us

$$\inf_{\Pi \in \mathcal{C}(\mu_0,\mu_1)} D_f(\Pi \| P) \leq \inf_{\pi \in \Pi(\mu_0,\mu_1)} D_f(\pi \| P_{01}) \tag{27}$$

Combining 21 and 27, we obtain:

$$\inf_{\Pi \in \mathcal{C}(\mu_0,\mu_1)} D_f(\Pi \| P) = \inf_{\pi \in \Pi(\mu_0,\mu_1)} D_f(\pi \| P_{01})$$

Having proved both directions, equality holds. $\qquad\square$

**Corollary 8.** *The optimal dynamic measure $\Pi^*$ is obtained by solving the static problem for $\pi_s^*$ and lifting it to the path space with conditional distribution matching the prior: $\Pi_{xy}^* = P_{xy} := P(\cdot|x,y)$.*

### A.2 PROOF OF THEOREM 2: MSSB-KD AND INFORMATION BOTTLENECK

*Proof.* We establish the connection between MSSB-KD and the Information Bottleneck principle through a detailed analysis of the objective's information-theoretic structure.

Consider the MSSB-KD formulation on three spaces: the data space $(\mathcal{X}, \mathcal{B}_{\mathcal{X}})$, teacher logit space $(\mathcal{T}, \mathcal{B}_{\mathcal{T}})$, and student logit space $(\mathcal{S}, \mathcal{B}_{\mathcal{S}})$, each equipped with their Borel $\sigma$-algebras. Let $\mu_{\mathcal{X}}$, $\mu_{\mathcal{T}}$, and $\mu_{\mathcal{S}}$ denote reference measures on these spaces, and let $\pi \in \mathcal{P}(\mathcal{X} \times \mathcal{T} \times \mathcal{S})$ be a joint probability measure. The MSSB-KD objective seeks to minimize:

$$J_{\text{MSSB-KD}}(\pi) = \int_{\mathcal{X} \times \mathcal{T} \times \mathcal{S}} \left(\|s-t\|^2 + \lambda_{\text{ce}} \mathcal{L}_{\text{CE}}(s,t)\right) d\pi(x,t,s) + \varepsilon \cdot \text{KL}(\pi \| \mu_{\mathcal{X}} \otimes \mu_{\mathcal{T}} \otimes \mu_{\mathcal{S}}) \tag{28}$$

The key insight lies in decomposing the KL divergence term. Through the chain rule for relative entropy, we can separate the divergence from the product reference measure into meaningful information-theoretic quantities. Denoting the marginals of $\pi$ as $\pi_X$, $\pi_T$, and $\pi_S$, we obtain:

$$\text{KL}(\pi \| \mu_{\mathcal{X}} \otimes \mu_{\mathcal{T}} \otimes \mu_{\mathcal{S}}) = \text{KL}(\pi \| \pi_X \otimes \pi_T \otimes \pi_S) \tag{29}$$

$$+ \text{KL}(\pi_X \| \mu_{\mathcal{X}}) + \text{KL}(\pi_T \| \mu_{\mathcal{T}}) + \text{KL}(\pi_S \| \mu_{\mathcal{S}}) \tag{30}$$

This decomposition reveals that the KL penalty consists of two components: the total correlation $\text{TC}(X; T; S) = \text{KL}(\pi \| \pi_X \otimes \pi_T \otimes \pi_S)$, which measures the total dependency among the three variables, and the marginal divergences, which regularize the individual distributions.

The total correlation itself admits multiple decompositions. For this analysis, the following form is particularly revealing because it explicitly isolates terms related to the Information Bottleneck's objectives of compression and fidelity. We use the identity:

$$\text{TC}(X; T; S) = I(X; S) + I(T; S) + I(X; T|S) \tag{31}$$

One can verify this identity by expanding the right-hand side (RHS) using fundamental entropy definitions.

$$
\begin{aligned}
\mathrm{RHS} &= I(X;S) + I(T;S) + I(X;T|S) \\
&= [H(X) + H(S) - H(X,S)] + [H(T) + H(S) - H(T,S)] \\
&\quad + [H(X,S) + H(T,S) - H(S) - H(X,T,S)] \\
&= H(X) + H(T) + H(S) - H(X,T,S) = \mathrm{TC}(X;T;S)
\end{aligned}
\tag{32}
$$

This decomposition is particularly revealing because $I(X;S)$ represents precisely the compression term that appears in the classical Information Bottleneck formulation. Since the total correlation is non-negative and includes $I(X;S)$ as an additive component, minimizing the KL divergence necessarily encourages compression—the student $S$ is incentivized to minimize its dependence on the input $X$.

The connection to the fidelity term requires a more subtle analysis. The cost function $\mathbb{E}_\pi[\|s - t\|^2 + \lambda_{\mathrm{ce}}\mathcal{L}_{\mathrm{CE}}(s,t)]$ measures the expected squared distance between student and teacher logits. Through the Donsker-Varadhan variational representation Donsker & Varadhan (1975), we can establish that minimizing this distortion is related to maximizing the mutual information $I(T;S)$. Specifically, for any joint distribution $\pi(t,s)$, we have:

$$
I(T;S) \geq -\mathbb{E}_\pi[\|s - t\|^2 + \lambda_{\mathrm{ce}}\mathcal{L}_{\mathrm{CE}}(s,t)] - \log \mathbb{E}_{\pi_T \otimes \pi_S}[e^{-\|s-t\|^2 + \lambda_{\mathrm{ce}}\mathcal{L}_{\mathrm{CE}}(s,t)}]
\tag{33}
$$

This bound shows that minimizing the expected squared error between teacher and student outputs is equivalent to maximizing a lower bound on their mutual information, which captures the fidelity objective of preserving teacher knowledge.

Combining these insights, the MSSB-KD objective can be understood as optimizing:

$$
J_{\mathrm{MSSB\text{-}KD}}(\pi) = \underbrace{\mathbb{E}_\pi[\|s - t\|^2 + \lambda_{\mathrm{ce}}\mathcal{L}_{\mathrm{CE}}(s,t)]}_{\text{Distortion penalty}} + \varepsilon \cdot \underbrace{\mathrm{TC}(X;T;S)}_{\text{Total correlation}} + \varepsilon \cdot \underbrace{\sum_{i \in \{X,T,S\}} \mathrm{KL}(\pi_i \| \mu_i)}_{\text{Marginal regularization}}
\tag{34}
$$

This formulation achieves three objectives simultaneously. First, it implements compression by penalizing $I(X;S)$ through the total correlation term. Second, it maintains fidelity by minimizing distortion, which relates to maximizing $I(T;S)$ (a term also penalized by the total correlation, creating a trade-off - however, the tradeoff parameter $\varepsilon$ can be adjusted). Third, it provides regularization through the marginal constraints, preventing overfitting to specific distributions.

The relationship to the classical Information Bottleneck becomes clearer when we consider their structural differences. The classical IB formulation assumes a Markov chain $T \leftarrow X \rightarrow S$ and optimizes $\min_{p(s|x)}[I(S;X) - \lambda I(S;T)]$, where $\lambda$ controls the compression-fidelity trade-off. In contrast, MSSB-KD operates on general joint distributions without imposing Markov structure, using squared error as a tractable proxy for mutual information, and incorporating total correlation rather than just pairwise dependencies.

Under certain conditions—specifically when the reference measures are uniform or Gaussian, the data approximately follows the Markov structure, and the average distortion is small—MSSB-KD approximates the classical IB objective with trade-off parameter $\lambda \approx 1/\varepsilon$. However, MSSB-KD provides a more general framework that doesn't require these assumptions, making it applicable to a broader class of knowledge distillation problems while maintaining the core information-theoretic principles of compression and fidelity.

The entropy parameter $\varepsilon$ emerges as the natural analogue of the IB trade-off parameter $\lambda$: high values of $\varepsilon$ emphasize compression by strongly penalizing correlations, while low values prioritize fidelity by allowing tighter coupling between teacher and student. This connection provides principled guidance for hyperparameter selection based on the desired balance between model compression and performance preservation. $\qquad\square$

### A.3 TRAINING AND INFERENCE ALGORITHMS

**Algorithm 1** Block-Stochastic Optimization of Dual Potentials

**Input:** $\mu_1, \mu_2, \mu_3$, cost $c(x, y, t)$, learning rate $\eta$, batch size $B$
**Initialize:** $\theta_1, \theta_2, \theta_3$
**while** not converged **do**
    Sample minibatch $(x_i, y_i, t_i)_{i=1}^{B} \sim \mu_1 \otimes \mu_2 \otimes \mu_3$
    Compute stochastic estimate of objective $\mathcal{L}(\theta)$ via Eq. (7)
    $\theta_1 \leftarrow \theta_1 - \eta \hat{\nabla}_{\theta_1} \mathcal{L}$
    $\theta_2 \leftarrow \theta_2 - \eta \hat{\nabla}_{\theta_2} \mathcal{L}$
    $\theta_3 \leftarrow \theta_3 - \eta \hat{\nabla}_{\theta_3} \mathcal{L}$
**end while**

**Algorithm 2** Unadjusted Langevin Algorithm

**Input:** input latent $x^*$, text latent $t^*$, initial $y$ latent $y^{(0)}$, potential $\beta(y)$, cost $c(x, y, t)$, temperature $\varepsilon$, step size $\eta$, steps $K$
**Initialize:** set $y^{(0)} = y$       ▷ warm start
**for** $k = 0$ **to** $K - 1$ **do**
    Gradient of log-unnormalized posterior:
$$g^{(k)} \leftarrow \hat{\nabla}_y \left[ \beta(y^{(k)}) - \tfrac{1}{\varepsilon} c(x^*, y^{(k)}, t^*) \right]$$
    Sample Gaussian noise $\xi^{(k)} \sim \mathcal{N}(0, I)$
    Langevin update:
$$y^{(k+1)} \leftarrow y^{(k)} + \eta g^{(k)} + \sqrt{2\eta} \xi^{(k)}$$
**end for**
**Return:** generated latent $y^* = y^{(K)}$
Decode new image: $\hat{I} = G(y^*)$

## A.4 Convergence Guarantees

**Theorem 9** (Convergence of Alternating Optimization). *Under Lipschitz assumptions on the potential networks and bounded cost function, the alternating optimization scheme converges to a stationary point of the MSSB-KD objective when using full-batch gradients. For stochastic minibatch optimization with learning rate $\eta_k = \mathcal{O}(1/\sqrt{k})$, the algorithm converges to a neighborhood of a stationary point.*

*Proof.* The convergence analysis follows from the block coordinate descent-ascent framework of Xu & Yin (2013) and Razaviyayn et al. (2013).

For the full-batch case, the MSSB-KD optimization alternates between:

1. Updating potentials $(\alpha, \beta, \gamma)$ while fixing student parameters $\theta_s$

2. Updating student parameters $\theta_s$ while fixing potentials

The dual problem for potentials is concave under our assumptions, while the student loss $\mathcal{L}_s(\theta_s; \alpha, \beta, \gamma)$ is smooth and bounded below. Each alternating step either maintains or improves the objective value. By the convergence theorem for block coordinate methods Tseng (2001), this guarantees convergence to a stationary point.

For the stochastic minibatch case, we have additional gradient noise. Following Ghadimi and Lan Ghadimi & Lan (2013), with unbiased gradient estimates $\tilde{\nabla} f$ satisfying $\mathbb{E}[\tilde{\nabla} f] = \nabla f$ and bounded variance $\mathbb{E}[\|\tilde{\nabla} f - \nabla f\|^2] \leq \sigma^2$, the iterates satisfy:

$$\mathbb{E}\left[ \min_{t \leq T} \|\nabla f(x_t)\|^2 \right] \leq \mathcal{O}\left( \frac{1}{\sqrt{T}} + \frac{\sigma^2}{T} \right) \tag{35}$$

This implies convergence to a neighborhood of size $\mathcal{O}(\sigma\sqrt{\eta})$ around a stationary point, where $\eta$ is the final learning rate. With diminishing learning rates $\eta_k = \mathcal{O}(1/\sqrt{k})$, we achieve convergence in expectation. $\square$

**Theorem 10** (Convergence Rate of Block-Coordinate Ascent). *For the dual problem with strongly convex regularization, the block-coordinate ascent algorithm converges to the global optimum at a rate of $\mathcal{O}(1/k)$ for cyclic updates and $\mathcal{O}((1 - \rho)^k)$ for randomized updates, where $\rho$ depends on the problem conditioning.*

*Proof.* The dual MSSB problem can be written as:

$$\max_{\alpha,\beta,\gamma} D(\alpha, \beta, \gamma) = \sum_i \mathbb{E}[\phi_i] - \varepsilon \log Z(\alpha, \beta, \gamma) \tag{36}$$

where $Z$ is the partition function and $\phi_i \in \{\alpha, \beta, \gamma\}$ are the potentials.

Following Beck and Tetruashvili Beck & Tetruashvili (2013b), for a block-separable convex optimization problem with $L$-smooth blocks and $\mu$-strongly convex regularization, cyclic block coordinate descent achieves:

$$D^* - D(\phi^{(k)}) \leq \frac{3nL\|\phi^{(0)} - \phi^*\|^2}{2k} \tag{37}$$

where $n$ is the number of blocks (3 in our case) and $\phi^*$ is the optimal solution.

For randomized block selection with uniform probability, following Richtárik and Takáč Richtárik & Takáč (2014), the expected convergence rate is:

$$\mathbb{E}[D^* - D(\phi^{(k)})] \leq (1 - \rho)^k [D^* - D(\phi^{(0)})] \tag{38}$$

where $\rho = \frac{\mu}{n(L+\mu)}$ is the condition-dependent rate.

The $\mathcal{O}(1/k)$ rate for cyclic updates follows directly from the first bound, while the linear rate for randomized updates requires the strong convexity from our entropy regularization. $\square$

## A.5 MAPPER ARCHITECTURE

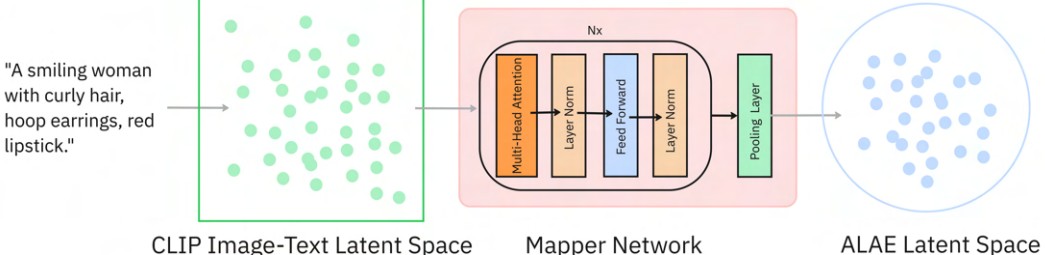

Figure 3: Cross-attention mapper architecture.

## A.6 IMPLEMENTATION DETAILS

**Backbones** We work in a $d{=}512$ latent space. Faces are encoded/decoded with a pretrained ALAE (FFHQ, $1024^2$). Text is encoded by CLIP ViT-B/16.

**Potentials** The dual potentials $(u, v, w)$ are 3-layer MLPs with hidden size 2048, LayerNorm + GELU between layers, and a final linear layer.

**Text $\rightarrow$ latent mapper** A frozen cross-attention mapper with 4 Transformer-style blocks, 16 heads, hidden dimension 2048, dropout 0.1. The Mapper Architecture is in A.5.

**Cost**

$$c(x, y, t) = \lambda_1 \|x - y\|_2^2 + \lambda_2 \|y - f_\theta(t)\|_2^2 + \lambda_3 \|y\|_2^2, \quad (\lambda_1, \lambda_2, \lambda_3) = (1.0,\ 3.0,\ 10^{-3}).$$

Training uses $\varepsilon{=}0.07$; at test time we report results for $\varepsilon \in [0.07, 0.15]$.

**Optimization** Adam with learning rate $1{\times}10^{-4}$, batch size 128, gradient-norm clipping at 1.0, 10,000 training steps, seed 2025.

**Sampling** ULA on $y$ with $K{=}5000$ steps, step size $5{\times}10^{-5}$;

## A.7 ADDITIONAL EXPERIMENTAL RESULTS

| text description | diverse generated images |
|---|---|
| An elderly man with white beard, narrow eyes, and thin lips. He wears a serious look. |  |
| A woman with long brown hair and brown eyes. She has a small nose and a smiling mouth. | |
| A man with short black hair, a strong jawline, and dark eyes. He has dark eyebrows and a serious expression. | |
| A middle-aged man with short dark hair and big ears. He has thick eyebrows and a bushy beard. | |
| A young man with curly dark hair, broad shoulders, and a confident stance. He looks focused and determined. | |
| A woman with blonde wavy hair and high cheekbones. She is wearing light makeup and small hoop earrings. | |
| A man with neatly combed brown hair, wearing rectangular glasses. He has thick eyebrows and a serious expression. | |
| She has short, halflength blonde hair and blue eyes. She looks serious. | |

Figure 4: **Text-to-image generation with MSSB (FFHQ).** Left: text descriptions. Right: diverse images sampled from our MSSB conditional plan for the same prompt (six independent seeds per row). The samples are high-quality and closely follow the descriptions; see Fig. 1 for Brownian-Bridge intermediates.

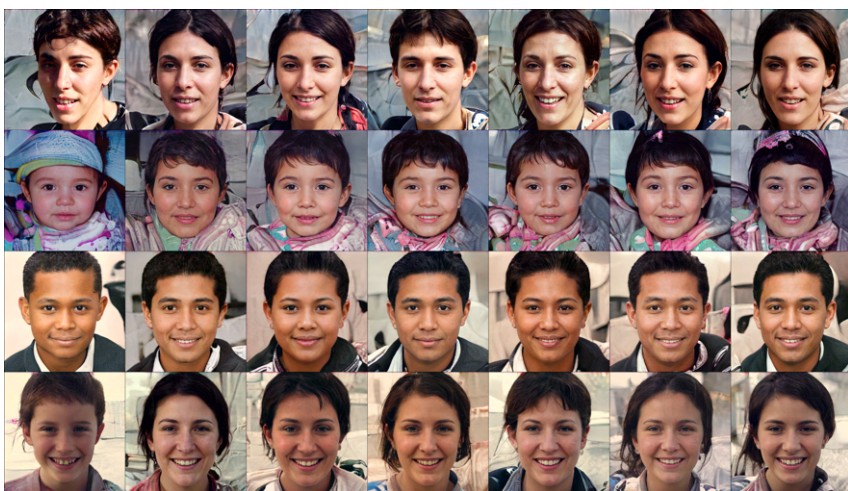

(a) **MSSB Children** → **Older**, $\varepsilon = 0.05$. Low diversity.

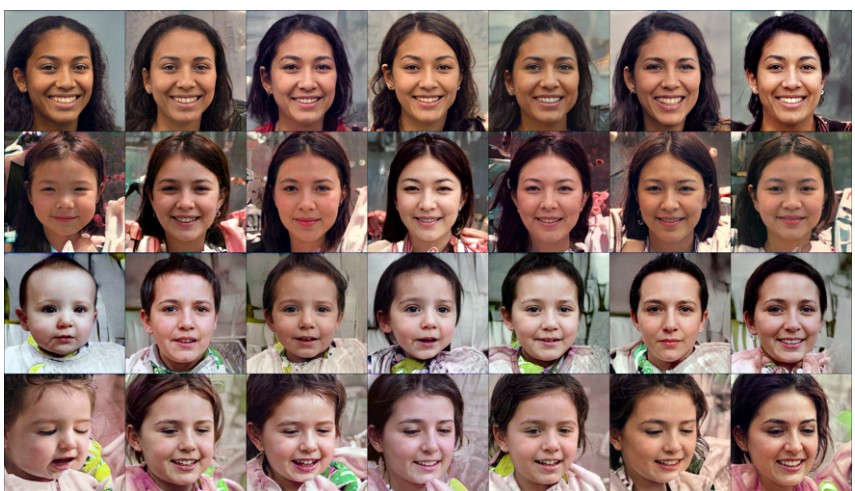

(b) **MSSB Children** → **Older**, $\varepsilon = 0.10$. Moderate diversity.

(c) **MSSB Children** → **Older**, $\varepsilon = 0.15$. High diversity.

Figure 5: Qualitative results for **MSSB Children** → **Older** with varying $\varepsilon$. Lower $\varepsilon$ yields more deterministic mappings; higher $\varepsilon$ increases stochasticity and visual diversity.

