# OpenReview forum: "Multi-Marginal f-Divergence Schrödinger Bridges: Towards a Unifying Framework for Generation and Distillation"
_ICLR.cc/2026/Conference — ICLR 2026 Conference Withdrawn Submission_

### Official Review · Reviewer_geRY · 2025-10-24

**Soundness:** 2
**Presentation:** 3
**Contribution:** 2
**Rating:** 4
**Confidence:** 3

**Summary:**

This paper introduces Multi-marginal Static Schrödinger Bridge (MSSB) with general f-divergence as a unified framework for multimodal tasks. Key contributions include proving equivalence between static and dynamic formulations beyond KL divergence, showing MSSB-based knowledge distillation implements a variational Information Bottleneck relaxation, and demonstrating strong empirical results in generation and distillation tasks with robustness to noise.

**Strengths:**

1. This paper makes several original contributions. Most notably, it extends the traditional two-marginal Schrödinger Bridge framework to handle three or more distributions.
2. The paper demonstrates certain theoretical and empirical quality.
3. The paper is well-structured and clearly written.

**Weaknesses:**

1. The paper lacks a critical analysis of computational scalability. While the authors demonstrate N=3 cases (image, text, output), real-world multimodal applications often require handling 4+ modalities (e.g., vision+language+audio+context). The paper doesn't address how their block optimization scales as N increases.
2. The generative modeling experiments focus exclusively on FFHQ, which has limited diversity compared to broader datasets like ImageNet or COCO. Similarly, KD evaluations use only CIFAR-10/100, omitting more challenging benchmarks like ImageNet or domain-specific tasks (medical imaging, satellite imagery).
3. The paper emphasizes computational advantages of SSB over DSB, but doesn't quantify the actual runtime/memory costs of their approach.

**Questions:**

See the Weaknesses section.

---

### Official Review · Reviewer_rdUR · 2025-10-29

**Soundness:** 2
**Presentation:** 3
**Contribution:** 3
**Rating:** 4
**Confidence:** 2

**Summary:**

The paper presents a framework based on Multi-Marginal f-Divergence Schrödinger Bridges, which aims to unify multimodal generation, image editing, and knowledge distillation under a single theoretical formulation. The approach extends Schrödinger Bridge theory to multiple marginals and arbitrary f-divergences, proposing applications in generative modeling and teacher–student distillation.

However, the niche of the paper is unclear. The advantages of MSSB over existing image generation or editing frameworks are not demonstrated. While the theoretical development is mathematically sound, the empirical validation is weak, lacking strong baselines, quantitative comparisons, and diverse datasets.

**Strengths:**

* The paper offers a theoretically interesting formulation connecting f-divergences, optimal transport, and multi-marginal Schrödinger Bridges.
* The topic is conceptually appealing and timely in theory-oriented machine learning research.

**Weaknesses:**

* **Unclear practical niche.** It remains ambiguous how this method fits into the broader landscape of generative modeling. The theoretical contribution is interesting, but the connection to practical tasks like generation or editing is underexplored.
* **Lack of demonstrated advantages over existing methods.** The paper does not clearly explain or empirically validate why the proposed approach is preferable to well-established image generation or editing techniques (e.g., diffusion-based). It remains unclear why practitioners or researchers should prefer this approach over well-established alternatives.
* **Missing comparisons and quantitative evaluation.** The paper lacks evaluation against relevant baselines and omits standard quantitative metrics such as FID, CLIP score. Only one dataset is used, which severely limits the robustness of the conclusions. Without quantitative analysis, it is difficult to gauge the practical value of the proposed framework.
* **Unconvincing image editing results.** The qualitative examples for editing, such as the “smiling” case, are weak and fail to demonstrate realistic or semantically consistent changes. The results appear minimal and do not substantiate the claimed improvement in editing control.

**Questions:**

* What concrete advantages does MSSB offer over existing diffusion or optimal transport–based generative models in terms of controllability or performance?

* Could the authors include quantitative comparisons on standard benchmarks using metrics such as FID or CLIP alignment?

* Does the framework generalize to more complex or multimodal domains?

* Can the authors provide stronger and more diverse editing examples that better illustrate the claimed benefits?

---

### Official Review · Reviewer_7vZt · 2025-10-29

**Soundness:** 2
**Presentation:** 3
**Contribution:** 2
**Rating:** 2
**Confidence:** 3

**Summary:**

The paper presents the framework for multi-marginal static Schrödinger Bridge (MSSB) problem to enable multimodal generation and knowledge distillation. MSSB is theoretically grounded and employs practical training with block-stochastic optimization and Langevin-based inference. MSSB is evaluated on text-to-image and text-guided editing tasks, and further extend it to knowledge distillation, demonstrating effective multi-teacher distillation and improved robustness to noisy teachers.

**Strengths:**

* The proposed framework for MSSB is well-presented and seem novel.

* The paper provides convincing theoretical results.

* The MSSB-KD demonstrates promising robustness to noisy teachers.

**Weaknesses:**

I believe the key problem of the paper is limited practical value and insufficient experimental evaluation:

* The paper lacks any quantitative results and comparisons to alternative approaches for text-to-image and editing tasks. Thus, it is difficult to assess the method in terms of fidelity and efficiency.

* Image editing results are inconsistent. The method often does not preserve the source images and does not perform the desired edit.

* MSSB-KD does not show performance gains over DKD on the clean data.

* For KD, the comparison with DOT[1] may be also valuable. This technique can consistently boost the performance of the KD baselines. Also, it is interesting if DOT may complement MSSB-KD as well.

**Minor**

* \cite needs to be replaced with \citep in many places
* In table 1, I guess "T-1" and "T-5" denote "top-1" and "top-5" accuracy, respectively. This can be clarified in the text.

[1] DOT: A Distillation-Oriented Trainer. ICCV2023

**Questions:**

Please address the concerns in Weaknesses

---

### Official Review · Reviewer_jjFu · 2025-10-31

**Soundness:** 2
**Presentation:** 2
**Contribution:** 1
**Rating:** 2
**Confidence:** 4

**Summary:**

The paper studies static multi-marginal Schrödinger Bridge problems with applications to different scenarios: **image editing / unpaired image2image** in small-dimensional latent spaces and **knowledge distillation** (in the context of classification problems). In case of the image2image problems, the authors suggest optimizing the dual objective with respect to the dual potentials. They perform translation via Langevin dynamics, applied to the optimal plan representation via potentials, and demonstrate applicability of the method in the latent space of ALAE autoencoder. In case of KD, they stick to the primal objective, establish its connection with the Information Bottleneck, and demonstrate robustness of the method to noise and high performance in multi-teacher settings.

**Strengths:**

* Theoretical part of the paper is well-organised, well-written, and easy to follow;

* Representation of the image editing problem via coupling the source domain, target domain, and text instructions in style of the reward-based sampling from latent space looks promising;

* The proposed KD method offers desirable properties: robustness to noise and high performance in the multi-teacher setting.

**Weaknesses:**

### Focus of the paper
The major concern that I have is that the paper does not have a clear focus. It claims contributions in different theoretical, methodological, and experimental aspects, which are almost disconnected (except being related to the multi-marginal SB). Moreover, each of these contributions feel limited. Overall, the paper feels more like a review of the potential applications of the multi-marginal SB formulation, rather than a focused methodological work. Despite having some promising ideas, I think the paper in the current form lacks substance for publication. It would greatly benefit from focusing on one of the more specific directions of research. Below I will try to summarize this observation and highlight this and other concerns in details.

### Theory
1) The authors outline $f$-divergence in the name of the paper and in the contributions. However, in the experiments with KD, the authors use standard KL divergence; in the I2I settings, the authors do not specify $f$ and do not ablate its choice. Thus, I feel like highlighting $f$ divergences here is not significantly grounded.
2) The authors highlight that they prove the equivalence between the static and dynamic $f$-divergence SBs. However, the dynamic formulation is not used anywhere in the methods (except Figure 1, where the (statically) generated pictures are interpolated with noise after generation, which may harm the paper's presentation by confusing the readers). Though I do not try to understate the result itself (since it clearly has potential in developing $f$-divergence analogues of the existing SB methods), the proof is a simple combination of the data-processing inequality for $f$-divergences with the chain-rule for the Radon-Nikodym derivatives, which does not seem to have enough substance on its own. Overall, I do not see the importance of these results inside the paper.

### Methodology
The authors propose two applications of the multi-marginal SB formulation: latent-space image editing, and knowledge distillation. However, they do not propose a unifying methodological framework and instead solve the dual problem for images and the primal problem for distillation, which negatively affects the interconnection between the parts of the work. Moreover,
1) It seems that the dual algorithm is a straight reformulation of the SCONES [1] algorithm for the multi-marginal setting. This significantly limits the contribution in terms of I2I applications. The corresponding reference is lacking in the manuscript;
2) I feel like the dual algorithm is very restrictive in terms of the latent space dimension and structure. It requires performing Langevin updates, which is computationally expensive and not very efficient in high-dimensional structured domains as e.g. clean images or latent representations in diffusion autoencoders;
3) The sampling scheme in the dual algorithm became tractable due to ignoring the $d \mu_2(y)$ part of the distribution, which corresponds to the unconditional density of the data (e.g. images or latents). This simplification was seemingly possible only due to the simple nature of the ALAE latent space. I would speculate that in case of the more complex distributions the algorithm would not work without the corresponding data score;
4) The authors formulate the prior distribution as $d R = \exp(\ldots) d\mu_1 d \mu_2 d \mu_3$, where the measures $\mu_i$ are coupled via the tensor product (e.g. correspond to independent variables). However, in the training process they correspond to the dependent images and text instruction. It is also not very clear how the text-image pairs are transformed into the triplets $(x, y, t)$;
5) The proposed algorithm for the KD problem is not clearly stated and lacks important details: how are the "total correlation" and "marginal regularization" calculated during optimization? Is there some restriction on the probabilistic model of $\pi$ by design, so the objective becomes tractable? The transition from the marginal-constrained to the unconstrained problem is also not clearly explained.

### Experiments
Some experimental results in the I2I section are not convincing. Some of the selected source pictures are very close to the target domain (e.g. smiling people in neutral $\to$ smiling). The translation often fails at fitting the target distribution.


[1] Score-based Generative Neural Networks for Large-Scale Optimal Transport

**Questions:**

1) Could you please tell how the triplets $(x, y, t)$ are constructed in the I2I setting?
2) It would be great if you could describe the algorithm for KD in more details: how is the plan $\pi$ parameterized, how are the regularization terms calculated during optimization?

---

### Note · Authors · 2025-11-24

**Comment:**

Thank you very much for the AC and the reviewers for their time and effort to evaluate our submission. We have decided to withdraw our submission.

**Withdrawal Confirmation:**

I have read and agree with the venue's withdrawal policy on behalf of myself and my co-authors.